# Characterizing the Countrywide Epidemic Spread of Influenza A(H1N1)pdm09 Virus in Kenya between 2009 and 2018

**DOI:** 10.3390/v13101956

**Published:** 2021-09-29

**Authors:** D. Collins Owuor, Zaydah R. de Laurent, Gilbert K. Kikwai, Lillian M. Mayieka, Melvin Ochieng, Nicola F. Müller, Nancy A. Otieno, Gideon O. Emukule, Elizabeth A. Hunsperger, Rebecca Garten, John R. Barnes, Sandra S. Chaves, D. James Nokes, Charles N. Agoti

**Affiliations:** 1Wellcome Trust Research Programme, Epidemiology and Demography Department, Kenya Medical Research Institute (KEMRI), Kilifi 230-80108, Kenya; ZDeLaurent@kemri-wellcome.org (Z.R.d.L.); jnokes@kemri-wellcome.org (D.J.N.); cnyaigoti@kemri-wellcome.org (C.N.A.); 2Kenya Medical Research Institute (KEMRI), Nairobi 54840-00200, Kenya; GKikwai@kemricdc.org (G.K.K.); LMayieka@kemri.org (L.M.M.); MOchieng@kemricdc.org (M.O.); Notieno@kemricdc.org (N.A.O.); 3Fred Hutchinson Cancer Research Center, Vaccine and Infectious Disease Division, Seattle, WA 98109, USA; nicola.felix.mueller@gmail.com; 4Centers for Disease Control and Prevention (CDC), Influenza Division, Nairobi 606-00621, Kenya; uyr9@cdc.gov (G.O.E.); Sandra.Chaves@sanofi.com (S.S.C.); 5Centers for Disease Control and Prevention, Division of Global Health Protection, Nairobi 606-00621, Kenya; enh4@cdc.gov; 6Centers for Disease Control and Prevention, Division of Global Health Protection, Atlanta, GA 30333, USA; 7Influenza Division, National Center for Immunization and Respiratory Diseases (NCIRD), Centers for Disease Control and Prevention, Atlanta, GA 30333, USA; dqy5@cdc.gov (R.G.); fzq9@cdc.gov (J.R.B.); 8School of Life Sciences and Zeeman Institute for Systems Biology and Infectious Disease Epidemiology Research (SBIDER), Coventry CV4 7AL, UK; 9School of Public Health and Human Sciences, Pwani University, Kilifi 195-80108, Kenya

**Keywords:** next-generation sequencing, hemagglutinin, phylodynamics, transmission

## Abstract

The spatiotemporal patterns of spread of influenza A(H1N1)pdm09 viruses on a countrywide scale are unclear in many tropical/subtropical regions mainly because spatiotemporally representative sequence data are lacking. We isolated, sequenced, and analyzed 383 A(H1N1)pdm09 viral genomes from hospitalized patients between 2009 and 2018 from seven locations across Kenya. Using these genomes and contemporaneously sampled global sequences, we characterized the spread of the virus in Kenya over several seasons using phylodynamic methods. The transmission dynamics of A(H1N1)pdm09 virus in Kenya were characterized by (i) multiple virus introductions into Kenya over the study period, although only a few of those introductions instigated local seasonal epidemics that then established local transmission clusters, (ii) persistence of transmission clusters over several epidemic seasons across the country, (iii) seasonal fluctuations in effective reproduction number (*R*_e_) associated with lower number of infections and seasonal fluctuations in relative genetic diversity after an initial rapid increase during the early pandemic phase, which broadly corresponded to epidemic peaks in the northern and southern hemispheres, (iv) high virus genetic diversity with greater frequency of seasonal fluctuations in 2009–2011 and 2018 and low virus genetic diversity with relatively weaker seasonal fluctuations in 2012–2017, and (v) virus spread across Kenya. Considerable influenza virus diversity circulated within Kenya, including persistent viral lineages that were unique to the country, which may have been capable of dissemination to other continents through a globally migrating virus population. Further knowledge of the viral lineages that circulate within understudied low-to-middle-income tropical and subtropical regions is required to understand the full diversity and global ecology of influenza viruses in humans and to inform vaccination strategies within these regions.

## 1. Introduction

The novel influenza A(H1N1)pdm09 virus strain emerged in North America during March–April 2009, spread rapidly among humans, and developed into the first human pandemic of the 21st century [1,2,3,4]. By July 2009, 168 countries reported a total of 162,300 laboratory-confirmed cases and over 1100 human deaths [5,6,7]. Subsequently, it was estimated that globally over 123,000 deaths from March to December 2009 were associated with A(H1N1)pdm09 virus infection [8]. The A(H1N1)pdm09 virus displaced the seasonal A(H1N1) virus and has continued to circulate as a seasonal virus in subsequent years, causing annual seasonal epidemics alongside influenza A(H3N2) and B viruses globally [9,10,11,12], including in Kenya [13,14,15,16,17].

The global surveillance of influenza viruses has resulted in the generation of an extensive collection of geographically and temporally comprehensive virus sequence data, which has provided an opportunity to explore the drivers of global spread of influenza viruses [18,19,20,21,22]. For instance, genomic analysis found that multiple independent introductions of genetically distinct A(H1N1)pdm09 virus lineages occurred in most countries; in the United Kingdom, only two of the many lineages that were introduced at the start of the pandemic were detected there 6 months later [9,23]. Elsewhere, a pair of A(H1N1)pdm09 virus transmission chains appeared to have persisted in west Africa for almost 2 years [24]. Since the A(H1N1)pdm09 virus became endemic, such spatiotemporal patterns of spread of A(H1N1)pdm09 viruses on a countrywide scale have, however, not been determined in many tropical/subtropical regions where virus circulation patterns do not show a clear seasonality as they do in temperate countries [25,26]. This is often due to insufficient spatiotemporally representative sequence data [27,28]. A more complete understanding of the regional and the global spread of influenza viruses, however, requires deeper and wider sampling from understudied tropical/subtropical regions [28].

Surveillance of influenza viruses is ongoing in several parts of Kenya [13,17,29,30]. These are carried out by the Kenya Medical Research Institute (KEMRI), Ministry of Health, Kenya, KEMRI—Wellcome Trust Research Program (KWTRP), and the USA Army Medical Research Directorate (USAMRD-K). In 2007, the Ministry of Health, Kenya with technical support from the Centers for Disease Control (CDC) Kenya country office (CDC-Kenya) established a National Influenza Surveillance System in response to the 2005 influenza A(H5N1) virus threat. The aims of the surveillance system were to identify circulating strains of influenza virus, describe the epidemiology and burden of influenza in Kenya, and serve as a component of an early warning system for pandemic influenza [13]. CDC-Kenya conducts surveillance for influenza and influenza-like illness (ILI) and severe acute respiratory illness (SARI) throughout the country; these are conducted in sentinel hospitals, health facilities at demographic surveillance sites, and refugee camps, which represent varied urban, rural, high mobility, and socioeconomic conditions [13]. The KWTRP Virus Epidemiology and Control (VEC) team established research collaboration with CDC-Kenya through a joint study and a countrywide pathway of transmission study entitled “Studies of the pathways of transmission of respiratory virus disease” (SPReD-Kenya)—http://virec-group.org/spred-kenya/ (accessed on 10 September 2021), which also provided samples for this study.

Influenza surveillance activities in Kenya reported at least four separate introductions of A(H1N1)pdm09 virus into the country during the pandemic in 2009, with the first laboratory-confirmed case reported on 29 June 2009 [31,32]. In an effort to fill the gap in studies that seek to describe the transmission of influenza viruses in tropical settings, we studied the Kenya-wide patterns of introduction and spread of A(H1N1)pdm09 virus since it was introduced into the local population in 2009. We isolated, sequenced, and analyzed 383 A(H1N1)pdm09 virus codon-complete genome sequences sampled between 2009 and 2018 from seven locations in Kenya alongside contemporaneously sampled global sequences, to investigate the introduction and spread of A(H1N1)pdm09 viruses to Kenya. 

## 2. Materials and Methods

### 2.1. Sample Sources and Molecular Screening

Samples analyzed in this study were collected through two health facility-based surveillance networks. The first involved continuous countrywide surveillance for influenza through SARI sentinel hospital reporting. This was undertaken at six sites supported by CDC-Kenya: Kenyatta National Hospital (KNH), Nakuru County and Referral Hospital (CRH), Nyeri CRH, Kakamega CRH, Siaya CRH, and Coast General Teaching and Referral Hospital (Appendix A) [13,14,16,33]. The second was the pediatric viral pneumonia surveillance undertaken at the Kilifi County Hospital (KCH) (Appendix A) [29].

In the CDC-supported surveillance sites, a total of 41,685 nasopharyngeal/oropharyngeal (NP/OP) swab samples were collected from SARI inpatient admissions of all ages from June 2009 through December 2018. Samples were stored in viral transport medium (VTM) at −80 °C prior to processing [13]. Of these, 41,102 swabs were tested for influenza A virus (IAV), with positive samples subsequently subtyped for A(H1N1)pdm09 and A(H3N2) viruses using real-time reverse transcription polymerase chain reaction (RT-PCR) [13,14,16]. A total of 1307 (3.2%) A(H1N1)pdm09 virus-positive samples were obtained. Of these, 418 (31.9%) A(H1N1)pdm09 virus=positive samples were selected for this analysis on the basis of an RT-PCR cycle threshold (Ct) of <35.0 (as a proxy for high viral load), an adequate sample volume for RNA extraction (>140 μL), and a balanced distribution of samples based on surveillance sites and years.

In the second facility-based surveillance undertaken at KCH from January 2009 through December 2018, a total of 6147 NP/OP swab samples were collected and tested from children <5 years admitted with syndromic severe or very severe pneumonia [15,29]. Samples were stored in VTM at −80 °C prior to molecular screening. Samples were screened for a range of respiratory viruses, including IAV, using a multiplex real-time reverse transcription (RT)-PCR assay employing Qiagen QuantiFast RT-PCR kit (Qiagen, Hilden, Germany) [34]. An (RT)-PCR Ct of <35.0 was used to define virus-positive samples [34]. A total of 157 (2.6%) IAV-positive specimens were identified from KCH; however, these were not subsequently subtyped for A(H1N1)pdm09 and A(H3N2) viruses [29]. Therefore, all IAV-positive specimens were utilized, and only the A(H1N1)pdm09 viruses were selected for the current analysis.

### 2.2. RNA Extraction and Multi-Segment Real-Time PCR (M-RTPCR) for IAV

Viral RNA extraction and M-RTPCR were conducted as previously described [35]. Briefly, viral nucleic acid extraction from IAV and A(H1N1)pdm09 virus-positive samples was performed using the QIAamp Viral RNA Mini Kit (Qiagen, Hilden, Germany). Ribonucleic acid (RNA) was reverse-transcribed, and the complete coding region of IAV genome was amplified in a single M-RTPCR using the Uni/Inf primer set [36] in 25 μL PCR reactions. Successful amplification was evaluated by running the products on 2% agarose gel and visualized on a UV transilluminator after staining with RedSafe Nucleic Acid Staining solution (iNtRON Biotechnology Inc., Seoul, South Korea).

### 2.3. IAV Next-Generation Sequencing (NGS) and Virus Genome Assembly

Following PCR, amplicons were purified, quantitated, and normalized to 0.2 ng/μL as previously described [35]. Briefly, following PCR, the amplicons were purified with 1× AMPure XP Beads (Beckman Coulter Inc., Brea, CA, USA), quantified with Quant-iT dsDNA High Sensitivity Assay (Invitrogen, Carlsbard, CA, USA), and normalized to 0.2 ng/μL. Indexed paired end libraries were generated from 2.5 μL of 0.2 ng/μL amplicon pool using a Nextera XT Sample Preparation Kit (Illumina, San Diego, CA, USA) following the manufacturer’s protocol. Amplified libraries were purified using 0.8× AMPure XP beads, quantitated using Quant-iT dsDNA High Sensitivity Assay (Invitrogen, Carlsbard, CA, USA), and evaluated for fragment size in the Agilent 2100 BioAnalyzer System using the Agilent High Sensitivity DNA Kit (Agilent Technologies, Santa Clara, CA, USA). Amplicon libraries were then diluted to 2 nM in preparation for pooling and denaturation for running on the Illumina MiSeq (Illumina, San Diego, CA, USA). Pooled libraries were denatured with sodium hydroxide, diluted to 12.5 pM, and sequenced on the Illumina MiSeq using 2 × 250 base pair (bp) paired-end reads with MiSeq v2 500 Cycle Kit (Illumina, San Diego, CA, USA). A 5% Phi-X (Illumina, San Diego, CA, USA) spike-in was added to the libraries to increase library diversity by creating a more diverse set of library clusters. Contiguous nucleotide sequence assembly from the sequence data was carried out using the FLU module of the Iterative Refinement Meta-Assembler (IRMA) [37] using IRMA default settings: median read Q-score filter of 30; minimum read length of 125; frequency threshold for insertion and deletion refinement of 0.25 and 0.6, respectively; Smith–Waterman mismatch penalty of 5; gap opening penalty of 10. All generated sequence data were deposited in the Global Initiative on Sharing All Influenza Data (GISAID) EpiFlu^TM^ database (https://platform.gisaid.org/epi3/cfrontend; accessed on 10 March 2021) using accession numbers listed in the metadata file in the report’s GitHub repository https://github.com/DCollinsOwuor/H1N1pdm09_Kenya_Phylodynamics/tree/main/Data/; accessed on 11 March 2021.

### 2.4. Phylogenetic Analysis 

Consensus nucleotide sequences were aligned using MUSCLE (https://www.ebi.ac.uk/Tools/msa/muscle/; accessed on 11 March 2021) and translated in AliView v1.26 [38], and the individual gene segments of Kenyan sequences were concatenated into codon-complete genomes using SequenceMatrix v1.7.8 [39]. Phylogenetic trees of A(H1N1)pdm09 virus genomes from Kenya and contemporaneously sampled global sequences were reconstructed with maximum-likelihood and bootstrap analysis of 1000 replicates. The best-fit nucleotide substitution models were inferred using IQ-TREE v1.6.11 [40,41] and those chosen by the Bayesian Information Criterion for each concatenated virus genome implemented. The phylogenetic trees were visualized and annotated using Figtree v1.4.4 (http://tree.bio.ed.ac.uk/software/figtree/; accessed on 12 March 2021) and ggtree [42]. The full-length hemagglutinin (HA) codon sequences of all viruses were used to characterize A(H1N1)pdm09 virus strains into genetic groups (i.e., clades, subclades, and subgroups) using Phylogenetic Clustering with Linear Integer Programming (PhyCLIP) [43] and the European CDC Guidelines (https://www.ecdc.europa.eu/en/seasonal-influenza/surveillance-and-disease-data/influenza-virus-characterisation; accessed on 12 March 2021). Representative reference sequences for genetic clade assignment were included. 

### 2.5. Estimating Population Dynamics of A(H1N1)pdm09 Virus in Kenya from Local Transmission Clusters

To analyze the introduction and local spread of A(H1N1)pdm09 virus in Kenya, we first differentiated between “transmission clusters” and “single introductions”. Transmission clusters were defined as groups of sequences that originated from a single introduction into Kenya. Ancestral state reconstruction of internal nodes based on Kenyan and global sequences was used to infer these transmission clusters using a maximum-likelihood approach in TreeTime [44]. To do so, all the sequences were assigned into two discrete location states: (i) Kenyan—all Kenyan sequences, and (ii) non-Kenyan—all sequences from anywhere else from the globe. All sequences that clustered together were considered to belong in the same transmission cluster if their common ancestral nodes were inferred to be in Kenya, whereas each individual transmission cluster was considered to be the result of a single introduction into Kenya. Clusters were named to reflect their placement within global genetic clades 1–7, for example, KENI-GC7 indicates Kenya cluster I viruses within global genetic clade 7. According to the identified transmission clusters, we utilized the metadata file and script available at https://github.com/DCollinsOwuor/H1N1pdm09_Kenya_Phylodynamics/tree/main/Analysis (accessed on 11 March 2021) to re-estimate the number of introductions in R programming software v4.0.2. The transmission clusters were then used to analyze the spread of A(H1N1)pdm09 virus in Kenya using two phylodynamic approaches. First, the effective reproduction number (*R*_e_), which is the average number of secondary cases generated by an infection, was estimated using a birth–death skyline (BDSKY) analysis [45], where all individual transmission clusters are assumed to be independent observations of the same process with the same parameters [46]. Second, the relative genetic diversity (effective virus population size over time) was estimated using Gaussian Markov random field (GMRF) coalescent smoothing of the effective population size [47].

### 2.6. Spatial Dynamics of A(H1N1)pdm09 Virus in Kenya 

We conducted phylogeographic analysis to assess virus spread among three geographical regions of Kenya: (i) central Kenya—Nairobi, Nakuru, and Nyeri, (ii) western Kenya—Siaya and Kakamega, and (iii) coastal Kenya—Mombasa and Kilifi (Appendix A) using methods implemented in BEAST v1.10.4 package [48]. The analysis was implemented with an asymmetrical discrete trait approach and applied the Bayesian stochastic search variable selection (BSSVS) model [49]. Phylogeographic inferences were visualized with the Spatial Phylogenetic Reconstruction of Evolutionary Dynamics using Data-Driven Documents (SPREAD3) software v0.9.7.1c [50]. To visualize the geographic spread of the virus over time, a D3 file was generated using SPREAD3 v0.9.7.1c. A Kenya geo.json file was used for visualization, and the resulting log files were used to calculate Bayes factor (BF) values for significant dispersal rates between discrete locations.

## 3. Results

### 3.1. IAV Sequencing and Genome Assembly

Among the 418 A(H1N1)pdm09 virus-positive samples that were available from the CDC-Kenya surveillance system, 414 (99.1%) passed pre-sequencing quality control checks and were loaded onto the MiSeq (Appendix A). This yielded 344 (83.1%) codon-complete and 66 (15.9%) partial A(H1N1)pdm09 virus genomes; four of the sequenced genomes were not successfully assembled due to a low number of IAV sequence reads. For this analysis, only the 344 codon-complete A(H1N1)pdm09 virus genome sequences were included. Among the 157 IAV-positive specimens available from KCH, 94 (59.9%) that passed pre-sequencing quality control checks were loaded onto the MiSeq; the 63 (40.1%) specimens that were not sequenced had inadequate sample volumes for RNA extraction (<140 μL) and failed pre-sequencing quality control checks (Appendix A). A total of 45 (47.9%) A(H1N1)pdm09 virus (39 codon-complete and six partial) and 49 (52.1%) A(H3N2) virus genomes (46 codon-complete and three partial) were successfully generated from sequencing. Only the 39 codon-complete A(H1N1)pdm09 virus sequences were included in these analyses (Appendix A).

### 3.2. Spatiotemporal Distribution of Sequenced Samples

The A(H1N1)pdm09 virus was detected throughout the study period in Kenya, with the number of observed cases fluctuating from year to year (Figure 1A). Different locations experienced epidemic peaks in different years. However, there were A(H1N1)pdm09 virus detections in all sites, except in mid-2010, 2012–2013, and 2016, when there was little transmission of A(H1N1)pdm09 virus throughout the different sites (Figure 1A). The proportion of sequenced samples roughly reflected the overall distribution of positives that were detected from each site (Figure 1A). All A(H1N1)pdm09 virus genetic groups were detected in most sites, with their majority detected in five or six of the seven sites, which suggests that these lineages were in circulation in most of the sites in Kenya without geographical restrictions during the study period (Figure 1B). Phylogenetic analysis showed that the sequenced 383 A(H1N1)pdm09 viruses from Kenya comprised seven genetic groups: clade 7 (*n* = 97, 25.3%), clade 6 (*n* = 132, 34.5%), subclade 6C (*n* = 10, 2.6%), subclade 6B (*n* = 47, 12.3%), subclade 6B.1 (*n* = 38, 9.9%), subgroup 6B.1A (*n* = 57, 14.9%), and subgroup 6B.1A1 (*n* = 2, 0.5%) (Figure 2). These detections varied by surveillance year: clade 7—2009–2012; clade 6—2009–2011; subclade 6C—2013–2014; subclade 6B—2014–2016; subgroup 6B.1—2015–2016; subgroup 6B.1A and subgroup 6B.1A1—2018 (Figure 1C).

### 3.3. Patterns of Introduction of A(H1N1)pdm09 Viruses into Kenya and Local Transmission Clusters

We first assessed how the 383 sequences from Kenya compared to 1587 sequences sampled from around the world between 2009 and 2018 (Appendix A—Collation of contemporaneous global sequence dataset) by inferring their phylogenies. The Kenyan genomes spanned the existing global diversity (Figure 3), which suggests exchange (most likely introductions into Kenya) of viruses with other areas around the globe. For example, phylogenetic tree trunk viruses predominantly originated from North America in 2009, consistent with the origins of A(H1N1)pdm09 virus in North America in 2009, which seeded global viruses in 2009–2010. Subsequently, Asia and Europe appeared to be the major source populations in 2010–2013 and 2014–2018, respectively (Figure 3). The three geographical regions also represent sources of introduction of A(H1N1)pdm09 virus into Kenya in 2009, 2010–2013, and 2014–2018. On the basis of a reconstruction of geographic ancestry, Kenyan sequences grouped into local transmission clusters within the global diversity (Figure 3). We inferred 30 transmission clusters (KENI-GC7 to KENXXX-GC6B.1A) (Figure 3), which suggests that the sampled sequences were the result of 30 independent introductions from areas outside of Kenya. However, despite the relatively strong clustering of Kenyan sequences into transmission clusters, there were relatively few (8/30) introductions that resulted in local epidemics. To investigate if this number was a strict lower bound for the introductions, we used random subsets of the 383 A(H1N1)pdm09 virus sequences from Kenya to re-estimate the number of introductions (Figure 4). We found that the number of estimated introductions started to flatten slightly with the number of sequences subsampled but still grew. This saturating relationship suggests that the study did not grossly underestimate the full number of A(H1N1)pdm09 virus variants in this report. Seven of the 30 transmission clusters appeared to have persisted over several epidemic seasons (Table 1), which provides evidence for multiyear persistence of individual A(H1N1)pdm09 virus transmission clusters in a specific locality. All seven transmission clusters that persisted between 2009 and 2018 consisted of viruses sampled across Kenya, which suggests that A(H1N1)pdm09 virus persistence in the country was not constrained geographically.

### 3.4. Estimating Population Dynamics of A(H1N1)pdm09 Virus in Kenya from the Local Transmission Clusters

We estimated the effective reproduction number to be between 1 and 1.5 throughout the study period when we quantified the amount of local transmission on the basis of local transmission clusters (Figure 5A). We inferred seasonal fluctuations in *R*_e_ between 2009 and 2018, with annual peaks in *R*_e_ usually occurring at the end of the year and annual drops in *R*_e_ following the annual peaks, with the estimated median often being below 1 (Figure 5A). The low *R*_e_ values throughout 2009–2018 are consistent with the occurrence of smaller epidemics in Kenya throughout the study period. Coalescent reconstruction of A(H1N1)pdm09 virus occurrence in Kenya revealed (i) seasonal fluctuations in relative genetic diversity after an initial rapid increase during the early pandemic phase, which broadly corresponded to epidemic peaks in the northern and southern hemispheres, (ii) higher genetic diversity (genetic diversity >10), with greater frequency of seasonal fluctuations observed during 2009–2011 and 2018, and (iii) lower genetic diversity (variation occurring between 1 and 10 in genetic diversity), with relatively weaker seasonal fluctuations sustained from 2012–2017 (Figure 5B).

### 3.5. Regional Spread of A(H1N1)pdm09 Virus in Kenya

To infer the patterns of spread of A(H1N1)pdm09 virus among three geographical regions of Kenya (central Kenya, western Kenya, and coastal Kenya), we summarized and visualized its geographic spread over time on the basis of significant rates of spread between the geographical regions. We observed significant rates of spread from western Kenya to central and coastal Kenya. Additionally, we observed supported rates of spread from coastal Kenya to western and central Kenya (Figure 6).

## 4. Discussion

We observed multiple A(H1N1)pdm09 virus introductions into Kenya over the study period, although only a few of those introductions instigated local seasonal epidemics that then established local transmission clusters, some of which persisted over several epidemic seasons across Kenya. Furthermore, we show that the seasonal epidemics were associated with a lower number of infections (low estimates of *R*_e_), consistent with estimates from other regions for A(H1N1)pdm09 virus during seasonal epidemics [51], as well as with seasonal fluctuations in virus genetic diversity. Additionally, the spread of A(H1N1)pdm09 virus in Kenya was characterized by countrywide transmission following virus introduction.

Genomic analysis of virus sequence data from Kenya during the pandemic in 2009 reported the introduction of clade 2 and clade 7 viruses in Kenya, although clade 2 viruses did not circulate beyond the introductory foci, while clade 7 viruses disseminated countrywide [52]. Here, through detailed genomic analysis, we extend these earlier observations to show that clade 7 and clade 6 viruses were introduced into Kenya during the pandemic, disseminated countrywide, and persisted across multiple epidemics in multiple locations as local transmission clusters. A key question in influenza virus evolution and epidemiology is whether viral lineages can persist at low levels of circulation on local and regional scales or whether new virus strains must be continually reseeded from a globally sustained gene pool [24]. The intensive sampling of viruses during the pandemic in 2009–2010 enabled the molecular epidemiology of IAV to be examined at such a high resolution that the introduction, persistence, and/or fade-out of individual transmission clusters in specific localities could be determined [5,22]. Our analysis revealed sustained persistence of seven A(H1N1)pdm09 virus transmission clusters for over 2 years, although increased sampling is required to confirm that isolates from other localities are not interspersed within these clusters.

Nonmolecular epidemiological studies have hinted at climate-driven patterns of influenza virus spread in Africa, for example, in Kenya [53] and Uganda [54], where climatic factors have been shown to influence the seasonality of influenza viruses. Therefore, persistence in such African countries might be facilitated by climatic variability, which can generate temporally overlapping epidemics in neighboring regions [24]. Such patterns have been associated with global migration and persistence of influenza viruses in East and Southeast Asia [19]. Our findings support a shifting metapopulation model of circulation of influenza viruses in which viruses may pass through any geographical region for a variable amount of time rather than perpetually circulate in fixed locations, whereby new virus strains can emerge in any geographical region, with the location of the source population changing regularly [19]. Wider and deeper sampling of viruses from understudied tropical and subtropical regions is, therefore, required for a more complete understanding of the regional and the global spread of influenza viruses. 

Inclusion of regional and global genome sequences deposited in GISAID significantly improved the power of our phylogenomic analyses, which showed that the Kenyan diversity was part of the global continuum. For example, we showed widespread mixing of Kenyan lineages with global viruses from Africa, Asia, Europe, North America, South America, and Oceania. The use of NGS technology to generate virus sequence data from Kenya enables further scrutiny of the available data to answer other key molecular epidemiological questions. For example, the sequencing depth achieved with NGS may allow for analysis of minority variant populations. Therefore, NGS may facilitate more focused selection of vaccine strains based on strains in circulation in specific regions, which may improve the effectiveness of vaccines.

The study had some limitations. Firstly, the samples analyzed in this study were collected from hospitalized patients with SARI (CDC-Kenya) or viral pneumonia (KCH). This strategy avoided NGS of samples from outpatient cases that may have been critical in reconstructing the patterns of introduction and spread of A(H1N1)pdm09 virus in Kenya. Secondly, the analysis in this report only involved the coding regions of the A(H1N1)pdm09 virus gene segments. Although noncoding regions are considered to be conserved, mutations that affect viral replication may occur, and this information may not have been captured in this study. Thirdly, the paucity of sequence data from other African countries limited the analysis of regional patterns of persistence of influenza viruses, since persistence may be facilitated by climatic variability that generates temporally overlapping epidemics in neighboring countries. Lastly, the prioritized samples were selected on the basis of anticipated probability of successful sequencing inferred from the sample’s viral load as indicated by the diagnosis Ct value. Such a strategy ultimately avoided NGS of some samples that may have been critical in reconstructing the patterns of spread of A(H1N1)pdm09 virus and persistence of transmission clusters.

In conclusion, although the intensity of influenza surveillance in Africa still lags behind that of other continents, our findings suggest that considerable influenza virus diversity circulates within the continent, including virus lineages that are unique to the region, as reported for Kenya; these lineages may be capable of dissemination to other continents through a globally migrating virus population. Further knowledge of the viral lineages that circulate within understudied tropical and subtropical regions is required to understand the full diversity and global ecology of influenza viruses in humans and to inform vaccination strategies within these regions.

## Figures and Tables

**Figure 1 viruses-13-01956-f001:**
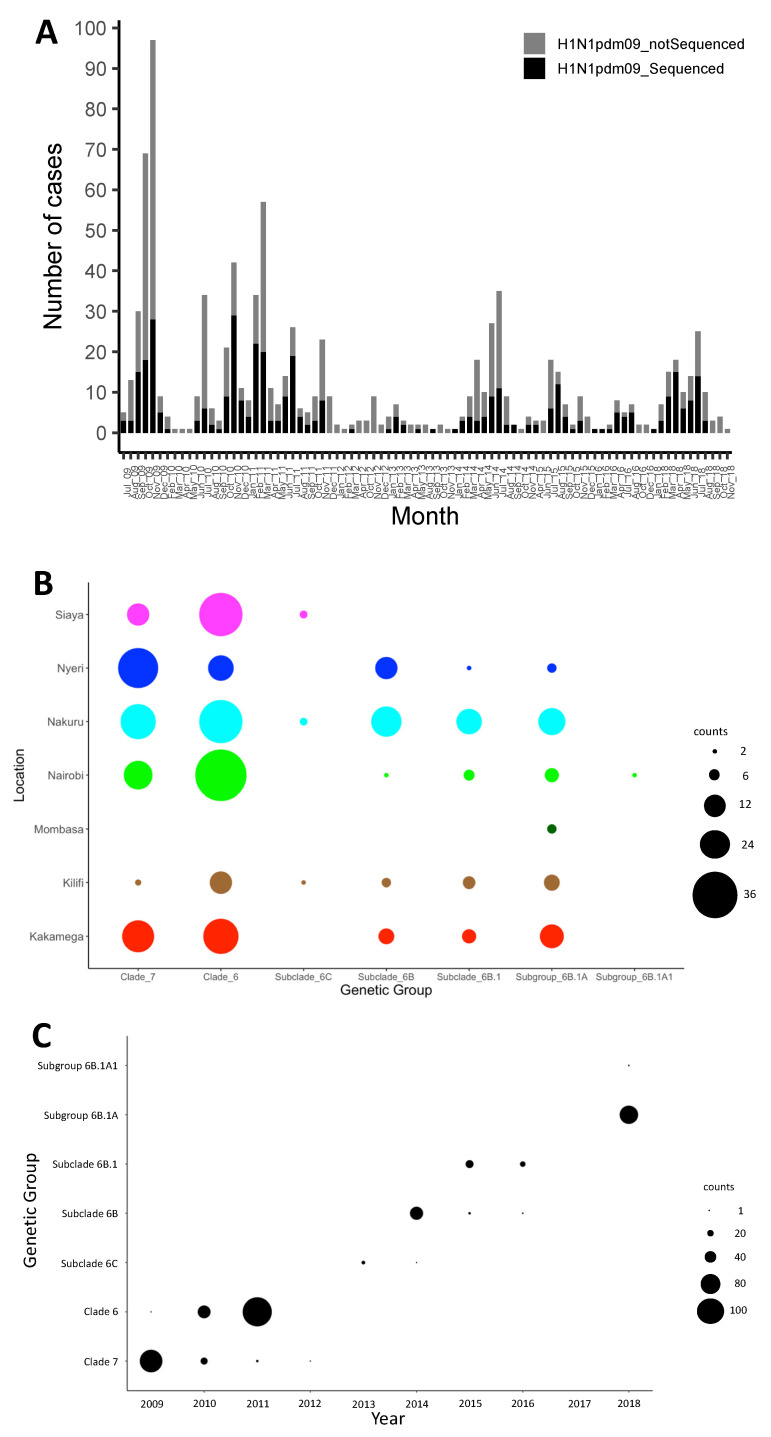
Distribution of influenza A(H1N1)pdm09 virus in Kenya, 2009–2018. (**A**) Bar plot showing number of A(H1N1)pdm09 virus-positive samples and sequenced positive samples by month between July 2009 and November 2018 in Kenya. All collected A(H1N1)pdm09 virus-positive samples and sequenced samples are indicated by color (all positive samples in gray—H1N1pdm09_notSequenced; sequenced samples in black—H1N1pdm09_Sequenced) as shown in the color key. (**B**) Bubble plot showing the distribution of genetic groups by location in Kenya, 2009–2018. (**C**) Distribution of genetic groups by surveillance year in Kenya, 2009–2018. The size of the circles in panels (**B**,**C**) is proportional to the number of samples, as shown in the counts key for the figures.

**Figure 2 viruses-13-01956-f002:**
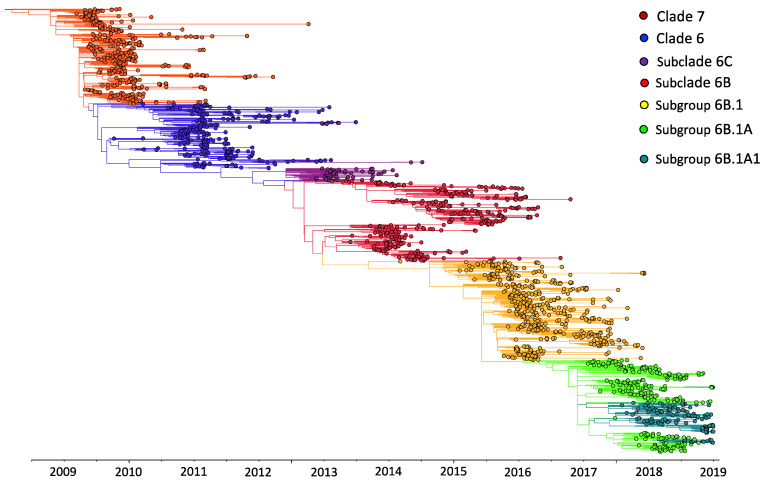
Maximum-likelihood phylogenetic tree of 1970 influenza A(H1N1)pdm09 virus sequences from Kenya and contemporaneously sampled global locations collected between 2009 and 2018. This is a time-calibrated phylogeny with time shown on the *x*-axis. Branches are colored on the basis of genetic group membership, as shown in the color key.

**Figure 3 viruses-13-01956-f003:**
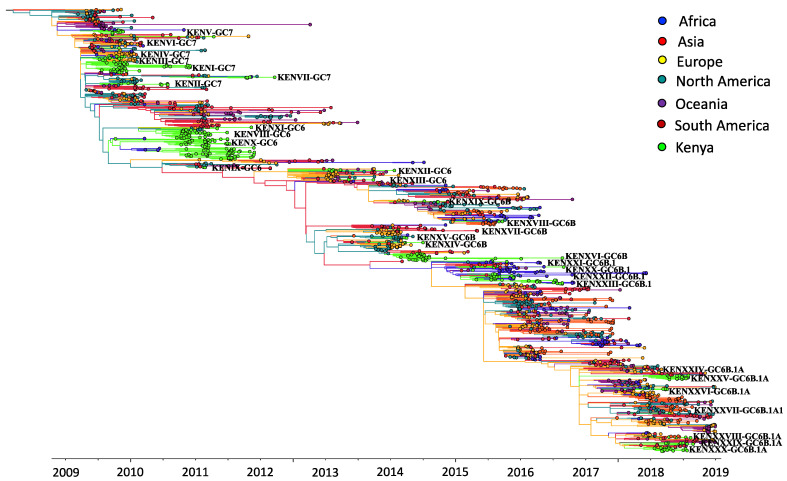
Time-resolved maximum-likelihood phylogenetic tree of Kenyan and contemporaneously sampled global sequences collected between 2009 and 2018 showing continent of sequence sampling and Kenyan transmission clusters. Unique Kenyan clusters are labeled with the prefix KEN, followed by cluster grouping and genetic group; for example, KENI-GC7 indicates Kenyan cluster I viruses, which fall within global genetic clade 7. The branches are colored on the basis of continent of sampling, as shown in the color key. Additionally, the trunk locations are inferred and colored by continent, which is based on geographic ancestry analyses of sampled sequences to indicate influenza A(H1N1)pdm09 virus origins.

**Figure 4 viruses-13-01956-f004:**
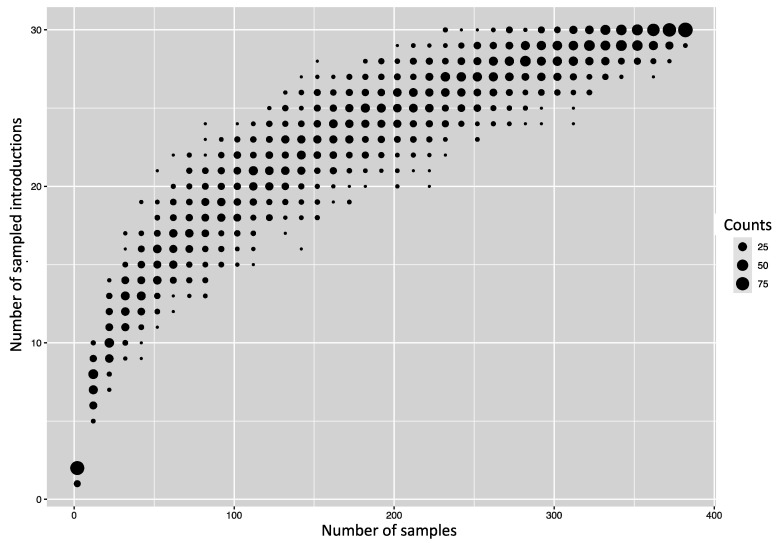
Number of introductions of A(H1N1)pdm09 viruses into Kenya depending on how many random sequences from Kenya were used to infer introductions.

**Figure 5 viruses-13-01956-f005:**
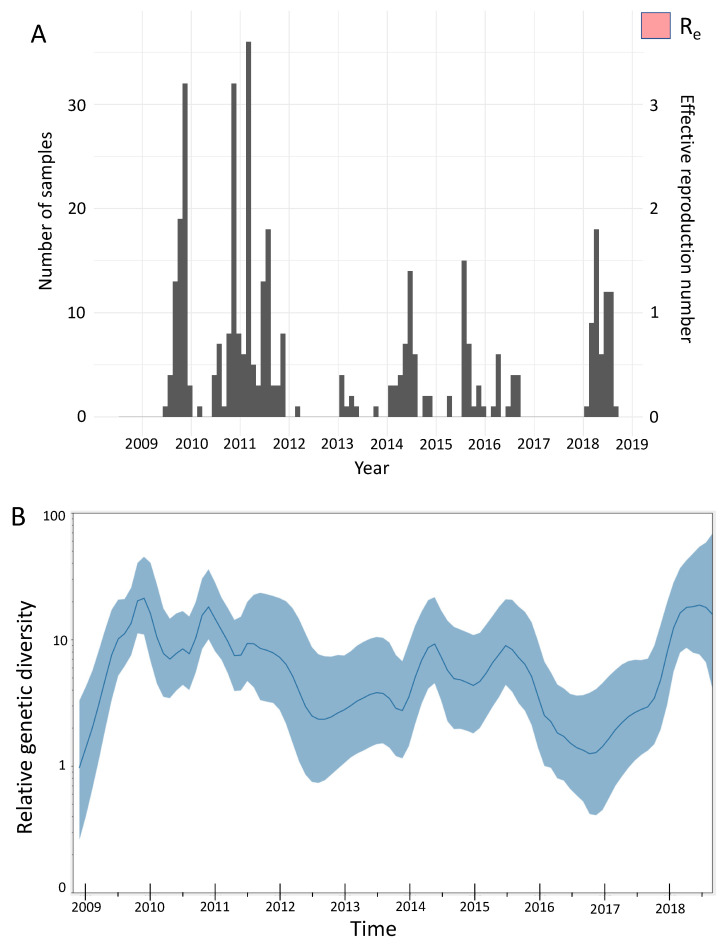
Population dynamics of influenza A(H1N1)pdm09 virus from Kenya, 2009–2018. (**A**) Estimates of the effective reproduction number through time inferred from all local clusters jointly by using BDSKY analysis. The primary *y*-axis shows the number of sequenced samples, while the secondary *y*-axis shows the effective reproduction number (*R*_e_). The dark pink section of the *R*_e_ values is the mean *R*_e_ estimate, whereas the light-pink margins denote the 95% confidence interval; time in years is shown on the *x*-axis. (**B**) Estimates of the relative genetic diversity through time for influenza A(H1N1)pdm09 virus from Kenya, 2009–2018, resolved using GMRF analysis. The dark-blue line is the mean estimate, and the blue margin denotes the 95% interval. The relative genetic diversity is shown on the *y*-axis, while time is shown on the *x*-axis. BDSKY, birth–death skyline; GMRF, Gaussian Markov random field.

**Figure 6 viruses-13-01956-f006:**
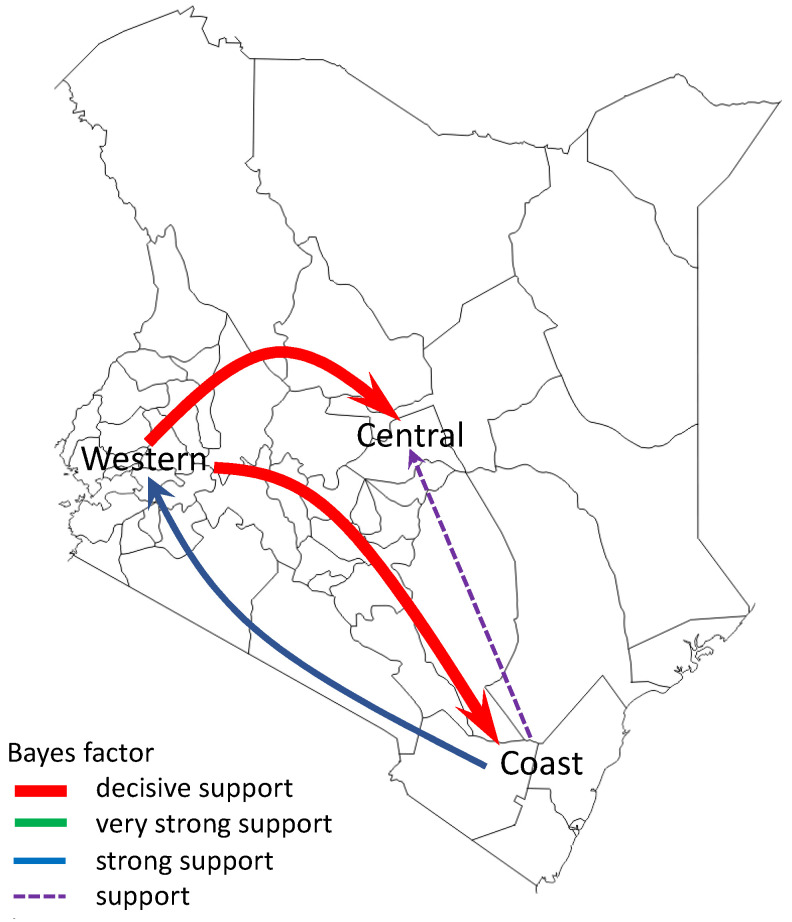
Networks of spread of influenza A(H1N1)pdm09 virus reconstructed using sequence data from Kenya, 2009–2018. Asymmetric pathways of spread between geographical regions of Kenya (central Kenya, western Kenya, and coastal Kenya) were inferred. Colored arrows indicate significant routes of spread from one location state to another, while line thickness represents the degree of statistical support. Red arrows are shown to indicate decisive routes of spread with Bayes factor (BF) support ≥1000; green lines represent very strongly supported routes with 100 ≤ BF < 1000; blue lines indicate strongly supported routes 10 ≤ BF < 100; purple dotted lines indicate supported routes with 3 ≤ BF < 10.

**Table 1 viruses-13-01956-t001:** Patterns of persistence of A(H1N1)pdm09 virus transmission clusters in Kenya.

† Cluster	Clade	Sequences	From	Circulation	Duration of Spread	Locations Detected
KENI-GC-7	7	49	Europe	July 2009 to December 2010	1.5 years	Nairobi; Nakuru; Nyeri; Siaya; Kilifi
KENVII-GC7	7	6	North America	December 2010 to March 2012	1.3 years	Nairobi; Nakuru; Kakamega
KENVIII-GC6	6	39	North America	October 2010 to June 2011	9 months	Nairobi; Nakuru; Kakamega; Nyeri; Siaya; Kilifi
KENX-GC6	6	88	North America	October 2009 to November 2011	2.2 years	Nairobi; Nakuru; Kakamega; Nyeri; Siaya; Kilifi
KENXII-GC6C	6C	8	Europe	January 2013 to January 2014	1 year	Nakuru; Siaya
KENXVI-GC6B	6B	34	North America	February 2014 to August 2016	2.5 years	Nairobi; Nakuru; Kakamega; Nyeri; Kilifi
KENXXIII-GC6B.1	6B.1	34	Europe	April 2015 to August 2016	1.3 years	Nairobi; Nakuru; Kakamega; Nyeri; Kilifi

† Cluster, name of transmission cluster; Clade, genetic group membership; Sequences, number of sequences in cluster; From, inferred geographical source of virus introduction; Circulation, duration of circulation of the cluster in Kenya; Locations Detected, Kenyan locations where clusters were detected.

## Data Availability

All generated sequence data were deposited in the GISAID EpiFlu^TM^ database (https://platform.gisaid.org/epi3/cfrontend; accessed on 11 March 2021) using accession numbers listed in the metadata file in the report’s GitHub repository https://github.com/DCollinsOwuor/H1N1pdm09_Kenya_Phylodynamics/tree/main/Data/; accessed on 11 March 2021.

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
