# Peer review of "Characterizing the Countrywide Epidemic Spread of Influenza A(H1N1)pdm09 Virus in Kenya between 2009 and 2018"

_viruses, 2021, doi:10.3390/v13101956_

Round 1

Reviewer 1 Report

The manuscript “ Characterizing the countrywide epidemic spread of influenza A(H1N1)pdm09 virus in Kenya between 2009 and 2018” has been reviewed.

The authors aim to describe the transmission of influenza A(H1N1)pdm09 viruses in Kenya as a simple of transmission in  tropical settings. Samples from postpandemic seasons from 2009 to2018 have been gathered and studied through phylogenetic sequence and compared to  global sequences, to investigate the introduction and spread of A(H1N1)pdm09 viruses into

Kenya.  Besides, a the study presents a phylogeographic analysis to assess virus spread among three regions of Kenya. Phylodynamics has become an increasingly popular statistical framework to extract evolutionary and epidemiological information from pathogen genomes. By harnessing such information, epidemiologists aim to shed light on the spatio-temporal patterns of spread and to test hypotheses about the underlying interaction of evolutionary and ecological dynamics in pathogen populations. In the actual context of the new pandemic caused by SARS CoV2 this framework is even more meaningful. Integrating various sources of information to deliver more precise insights in infectious diseases by connecting sequence to covariates of evolutionary and epidemic processes. Bayesian approach to covariate modeling and testing can generate insights into sequence evolution, trait evolution, and population dynamics in pathogen populations.

In all, this is a well written paper that is aceptable for publicaction . I would only suggest that because the study sample description includes only SARI and pneumonia cases with severe disease requiring hospital admission.  A brief description of the influenza surveillance system in Kenya would be informative to include in the introduction

Author Response

We thank the reviewer for reviewing our work. We are grateful for the recognition of the importance of our work in the context of the ongoing pandemic caused by SARS-CoV-2 virus and the application of our work in understanding the transmission dynamics of influenza viruses in understudied tropical and subtropical regions. We aimed to integrate various sources of information to deliver more precise insights in infectious diseases by connecting virus sequence data to covariates of evolutionary and epidemic processes, which the reviewer has recognized and applauded.

As recommended by the reviewer, we have included a paragraph in the introduction that describes the ongoing influenza surveillance system in Kenya; these are carried out by the Kenya Medical Research Institute (KEMRI), Ministry of Health, Kenya, KEMRI – Wellcome Trust Research Programme (KWTRP), and the USA Army Medical Research Directorate (USAMRD-K). The surveillance in Kenya is conducted in sentinel hospitals, health facilities at demographic surveillance sites, and refugee camps, which represent varied urban, rural, high mobility, and socio-economic conditions.

Reviewer 2 Report

A very well planned, conducted, and written study on the genomic evolution of H1N1 influenza in Kenya from 2008-2019. The authors provide important information and analysis on an understudied topic, the phylodynamics and phylogeography of influenza virus in tropical and sub-tropical regions. Our understanding of influenza virus evolution depends on high quality studies of global evolutionary dynamics and this article helps to fill an important knowledge gap. The authors are working in tandem with two surveillance studies, first, a countrywide sentinel surveillance for SARI, distributed in distinct geographical regions of the country and the second, an ongoing study of pediatric viral pneumonia. The access to samples from these surveillance studies, combined with state of the art sequencing gave the authors a beautiful data set to examine using sophisticated phylogenetic analyses.

Importantly, the authors demonstrated that viral dynamics included both many introductions into Kenya from all over the world and the persistence of viral lineages over multiple seasons during 2009-2018.  Interestingly, these persistent strains were always replaced by imported lineages, important information in assessing source/sink models of influenza evolution.

Overall, I recommend publication of this important work with minor revisions as listed below:

  1. p. 3 2.3. IAV next-generation sequencing (NGS) and virus genome assembly. Although mentioned in the results, the authors do not mention which platform and protocol are used for sequencing in the materials and methods, which seems an appropriate place.
  2. p.7 “To investigate if this number was a strict lower bound for the introductions, we used random subsets of the 383 A(H1N1) pdm09 virus sequences from Kenya to reestimate the number of introductions, Figure 4.” It is unclear to me, despite multiple back and forths between the results and the materials and methods, what methods were used to provide the analysis in Figure 4. Please add additional text to specify what methods were used.

Author Response

We are grateful for the review, which recognizes that our work was well planned, conducted, and written. We also thank the reviewer for recognizing that our work provides important information and analysis on an understudied topic based on the phylodynamics and phylogeography of influenza virus in tropical and subtropical regions.

We have addressed the two suggestions from the reviewer. Firstly, we have revised the next generation sequencing and virus genome assembly method to include the platform and detailed protocol that we used for sequencing in the materials and methods section. Secondly, we have described the method for resampling of Kenyan virus sequences, which we used to reestimate the number of introductions into Kenya using an R programming language script and a metadata file consisting of the identified clusters from Kenya.